# Health Professionals’ Perceptions of Pacific Co-Designed Resources for Pacific Gout Patients

**DOI:** 10.3390/healthcare13172089

**Published:** 2025-08-22

**Authors:** Samuela ‘Ofanoa, Malakai ‘Ofanoa, Siobhan Tu’akoi, Melenaite Tohi, Maryann Heather, Hinamaha Lutui, Rose Lamont, Felicity Goodyear-Smith

**Affiliations:** 1Pacific Health Department, School of Population Health, University of Auckland, Auckland 1023, New Zealand; 2Southpoint Family Doctors, Auckland 2104, New Zealand; 3Te Huinga Raukura ki Manurewa, Auckland 2102, New Zealand; 4Department of General Practice & Primary Health Care, School of Population Health, University of Auckland, Auckland 1023, New Zealand

**Keywords:** health professionals, health education, health resource, Pacific, gout, arthritis, co-design, feasibility

## Abstract

Background/Objectives: Pacific peoples in Aotearoa, New Zealand experience the highest burden of gout globally, yet there is still a lack of awareness and understanding of the disease. A Pacific community group and Pacific health professional network co-designed Pacific gout resources to improve understanding. The aim of this study is to identify and discuss the current state and perceptions of Pacific gout education, and explore health professionals’ views on Pacific co-designed resources and their usefulness in clinical settings. Methods: The Fa’afaletui model was utilised to conduct semi-structured Talanga interviews with 14 health professionals in Auckland, New Zealand who work in primary care clinics. The interview explored their views on providing gout education and on the feasibility of the Pacific co-designed gout resources. Talanga interviews were audio recorded and thematically analysed. Results: Overall, health professionals responded positively to the co-designed resources, identifying the benefits of supporting primary care consultations and improving Pacific patients’ understanding of gout. The key findings were summarised in five main themes: (1) health system barriers to gout education, (2) misleading information, (3) health professionals’ experiences of providing health education, (4) general impressions of Pacific co-designed resources, and (5) the feasibility of Pacific co-designed educational resources in a clinical setting. Conclusions: This study presents the views of health professionals in providing health education related to gout and on the feasibility of Pacific co-designed educational resources. It reinforces the significance of involving communities in the design and implementation of interventions to ensure they are culturally safe, relevant, and have long-term impacts on gout management.

## 1. Introduction

Education plays a vital role in improving the health and wellbeing of individuals by providing increased access to employment, economic opportunities, and social support [1]. Health education and literacy can be supported by resources that equip individuals with the knowledge and competencies to understand information and navigate services, empowering them to take control of their lifestyle choices [2]. To be effective, health resources must be developed carefully, taking into account comprehension levels and the type of format (visual, written, and engaging) through which the information is presented [3].

While many health educational resources have been developed in response to diverse health issues, there is a need to incorporate patient perspectives and experiences in the design phase [4]. A randomised control study in the United Kingdom of 132 patients aged 18 years or older reported that participants receiving an education intervention programme, which was designed based on patients’ prioritised needs, significantly improved self-efficacy and health outcomes [5]. Resources designed based on patients’ preferences and needs will motivate them to engage more with the resources, adopting changes in their behaviours and improving their adherence to treatment and self-management.

Gout is a painful inflammatory arthritis caused by a buildup of uric acid in the blood that deposits as crystals in the joints, resulting in gout flares [6]. If left untreated, it can cause joint damage and other comorbidities, such as kidney and cardiovascular disease [7,8]. In 2017, 41 million people were diagnosed with gout worldwide and each year, there are approximately 7 million newly diagnosed. This equates to about one million years lived with disability because of gout. Despite the high burden of gout and the availability of effective gout treatments, gout is still not adequately managed. Education is a strong predictor of improved adherence to gout urate-lowering therapy (ULT) [9]. However, in today’s society, people have access to an extensive array of information from various platforms such as the internet, legacy media, and social media. A content analysis by Dyuck et al. [10] highlighted that popular newspapers portray gout as a self-inflicted disease, focusing on its association with diet and linking it to social embarrassment and humour. Without appropriate regulations and scientific guidelines, these sources of information can often be misleading, impacting patients’ health-seeking behaviours for gout [10,11,12].

In Aotearoa New Zealand (NZ), gout affects six percent of those aged 20 years or older [13]. Pacific peoples experience the highest burden of gout, with a prevalence of 14.8% compared to 4% in non-Māori, non-Pacific peoples [13]. Māori gout prevalence falls within this, with 8.5% [13]. Furthermore, Pacific peoples are diagnosed with gout at a younger age and have higher rates of recurrent flares and gout-related hospitalisation [14]. Although Pacific peoples experience these inequities in gout health, they are less likely to receive effective gout management treatments [14,15]. A scoping review of gout interventions in NZ reported that although there were diverse educational resources available, these did not translate into changes in health behaviours. The study highlighted that communities needed to be consulted earlier and be decision-makers during all phases of resource development [16].

This paper presents research findings that started from two groups, the Pacific Peoples Health Advisory Group (PPHAG) of community members and the Pacific Practice-Based Research Network (PPBRN), with clinical staff from South Auckland general practices. Through workshops, they identified gout as a priority area that needed more innovative, co-designed approaches for Pacific communities [16,17]. These community-based groups also highlighted that there was still a lack of awareness and understanding about gout in Pacific peoples, particularly regarding risk factors and medications [18]. They advised that health education resources for Pacific gout patients needed to be simpler, more visual, and relevant [16]. Through the principles of co-design, the PPHAG and PPBRN, in collaboration with University of Auckland researchers, co-created two Pacific gout educational resources in the form of a video and a brochure. Although health professionals (PPBRN) were involved in the brainstorming and development of these resources, we wanted to explore the broader views of Pacific health professionals working in predominantly Pacific general practice clinics. Therefore, the present study has two main aims: (1) to identify and discuss the current state of and views on Pacific gout education within primary care practice in NZ, and (2) to explore health professionals’ perceptions of Pacific co-designed educational resources and their usefulness in clinical settings.

## 2. Materials and Methods

Ethical approval for this study was obtained from the Health and Disability Ethics Committee on 12 August 2024 for three years, Reference number HDEC20343.

### 2.1. Study Design

We adopted the Samoan Fa’afaletui model as the underlying framework for driving the research procedures of this study. Fa’afaletui upholds the Pacific cultural values of respect, relationships to weave together diverse perspectives, and knowledge to achieve a consensus in a culturally appropriate way [19,20]. An interpretivist approach was also utilised to investigate how health professionals subjectively understand gout education, drawing on their experiences and understanding from treating and engaging with Pacific patients [21,22]. This approach also takes into consideration that the cultural and historical contexts in which these health professionals are situated may shape their perspectives [22,23]. The consolidated criteria for reporting qualitative research (COREQ) was utilised to maintain the quality reporting of the study findings (Appendix A) [24,25].

### 2.2. Participants and Sampling

A purposive snowball sampling strategy was used to recruit Pacific health professionals for this study. The eligible criteria were health professionals such as general practitioners (GP), nurses, pharmacists, and health coaches who work in primary care settings in Auckland with a high proportion of Pacific patients. The initial participants were identified through the research team’s established networks, such as Pacific health organisations and community leaders, who were contacted via email and phone. Subsequently, a snowball sampling technique was implemented, whereby existing participants were invited to refer other eligible individuals for inclusion in the study. All participants who were invited were willing to participate, resulting in no dropouts. Data saturation was reached at 12 and a further two interviews were conducted to ascertain that no new information surfaced. The final sample size was 14 participants [26].

### 2.3. Data Collection

Semi-structured interviews were conducted with eligible participants from February to June 2025, adopting the Tongan Talanga methodology of interactive talk with a purpose [27]. All interviews were delivered via Zoom at a time and date chosen by the participants. Each interview session was audio recorded after receiving consent from the participant. The first two interviews were conducted by the Pacific researchers M.T., S.O., and M.O., as a pilot, informing revisions before M.T. completed the remaining sessions. The researchers took notes during the interviews and gave the participants a summary at the end. This enabled an opportunity for participants to clarify whether the researcher had missed or misinterpreted any important areas.

Each Talanga interview was conducted over a period between 40 and 60 min. In recognition of the participants’ time and contribution, they were gifted an NZD 100 grocery voucher at the end of the session. The main goal of the interview was to enable participants to discuss their views on the current state of gout education among Pacific communities. Therefore, the interview guide consisted of four key areas: the practitioner’s background information, current views on Pacific patient gout education in their practice, views on a community co-designed educational resource, and their feasibility and acceptability in a clinical setting (Appendix A).

### 2.4. Data Analysis

Each interview was transcribed by M.T. verbatim immediately after the interviews were completed and then imported into the NVivo software. Each transcript was labelled with a unique participant number, ethnicity, type of health professional, and age. Reflexive thematic analysis by Braun and Clarke [28,29] was adopted to inductively analyse the raw data, identifying patterns, relationships, and themes [29]. This process encourages a more active, reflexive role of the researchers when collecting and analysing data, particularly with all researchers collecting and analysing data being of Pacific ethnic descent [29]. Thematic analysis involved the following six steps: (1) familiarisation with the data, (2) generating initial codes, (3) constructing themes, (4) reviewing the themes, (5) defining the themes, and (6) writing the results [28,29]. By utilising these steps, data from all the interviews were initially coded independently by the Pacific researchers S.O. and S.T. Then, both researchers compared the theme interpretations and coding. Any discrepancies and similarities in their analysis were discussed in a brainstorming session to gain a more holistic and in-depth understanding of the results. The themes were then presented to the remaining researchers in the team (M.O., F.G., H.L., M.H., and R.L.) for review before progressing with the write-up of the results. Quotations were identified that were representative of the main themes identified in the findings.

## 3. Results

The mean age of the 14 participants was approximately 38 years (SD 7.0) and the sample was predominantly made up of females (64%) compared to males (36%). About 43% were Tongan, followed by Samoan and Cook Islands Māori (29% and 21%, respectively), with a small percentage of NZ European origin (7%). The majority of health occupations were nurses (50%), followed by GPs (36%) and pharmacists (14%)—see Table 1.

All participants expressed concern about gout due to witnessing its high prevalence in Pacific patients at their clinics or working environments and communities. Some health professionals also shared experiences of their family and seeing patients’ family members affected by gout.

The findings from this study revealed five main themes: (1) health system barriers inhibiting quality gout education, (2) misleading information, (3) health professionals’ experiences of providing health education, (4) general impressions of Pacific co-designed resources, and (5) perceptions of the feasibility of the resources in a clinical setting.

### 3.1. Theme 1: Health System Barriers Inhibiting Gout Education

The majority of participants (*n* = 8) highlighted barriers within the healthcare system that contributed to persistent gaps in understanding for Pacific gout patients [Table 1]. Common issues identified included access to the health system (difficulty securing appointments, waiting time, or cost), short consultation times (referring to 15 min set time for primary care appointments in NZ), lack of resources, and limited health professional knowledge. These factors hampered the ability of health professionals to provide quality gout education, impacting patients’ experiences and trust in the health system. One community pharmacist indicated the following:


*“A lot of people, they tell us they don’t really have time to see a doctor, when they book in to see a doctor they might not be able to get an appointment for like a week or so but they want something now, and when the pain’s gone, and a lot of them don’t end up really going to those appointments”*
*[P10]*.


*“We only get 15 minutes, 15 minutes does nothing! I am still trying to build rapport in 15 minutes”*
*[P6]*.


*“To be honest, I think I don’t even know if we’ve even got a pamphlet. I don’t even think we’ve got anything”*
*[P8]*.

Health professionals also discussed how the distribution of resources within clinics frequently changes based on the priorities communicated by higher authorities. However, when new health concerns arise, the staff and resources are redirected to these more pressing areas, resulting in the inconsistent delivery of health education. One GP stated the following:


*“Depending on what’s coming out from [refers to NZ Primary Health Organisation] to the GP clinics like there’s a rise in this, there is a rise in that, we have to shift our focus, and they will start implementing interventions like [NZ Pharmacy-led gout intervention in collaboration with nurses and GPs]. In our teams, once I am finished with my gout patient, then they have time with the nurse to provide more education about gout. There was a time where that was happening but slowly, once we get things back onboard, it starts to drift off again and I lose that resource. So, they are like, okay now we need the nurse to run whatever is peaking now and so the focus shifts to something else, so we drop again.”*
*[P6]*.

Four participants highlighted that they knew more about other health issues than gout, which impacted their ability to provide effective gout education. One nurse stated the following:


*“… [my knowledge of gout] was very little compared to when we say we talk about diabetes, a lot of our health professionals literally know the ins and out of diabetes, but gout is a very difficult one when we’re having to do a lot of education”*
*[P12]*.

### 3.2. Theme 2: Misleading Information Impacting Understanding of Gout and Stigma

For some health professionals (*n* = 5), particularly nurses, the main focus of their discussions with gout patients was diet, which can be misleading. Participants highlighted the need to contextualise dietary advice to avoid self-blame, feeling “fakamā” (embarrassment) [P9] and stigma. Two GPs reflected that while gout was often viewed as funny and joked about within Pacific families, it could exacerbate stigma and shame.


*“It is hard for them to turn their lifestyle around. When we tell them that food causes gout, it is really hard for [them] to turn around from eating the normal food that they usually eat”*
*[P5]*.


*“Not knowing that gout, 80% or 90% of it is more genetics and I tell them that when they come into my room it’s not their fault, and it’s nothing that they’re doing. The food that you’re eating, is a part of [contributing to] gout, but it’s a very small percentage of it”*
*[P6]*.


*“A lot of people know about gout because it’s one of those funny things like people will walk in limping and say oh, I’ve got the gout again, or, like their family members will be laughing at them when you say, oh, I think this might be gout”*
*[P14]*.


*“We joke around about it, but they think it’s their fault and that’s why they act the way they do, they’re quite secretive about it”*
*[P6]*.

Two nurses discussed the role of social media and family experiences of treating gout in promoting misleading information, particularly around the use of herbal remedies. One nurse stated the following:


*“…on social media, there are numerous promotions of all sorts of herbal medicine so people will turn and go take them just from looking at others [family and friends], oh that person is better from taking that herbal medicine so I will try that out too”*
*[P4]*.


*“Patients may view that herbal is better. Some people, they get their knowledge from other people. This will influence other people to think that the other health complications with the gout medicine is due to taking the gout medicine every day. People will then think to change and take the herbal. They can also think that herbal will cure them better. Like herbal is the shortcut to getting their gout fixed”*
*[P5]*.

### 3.3. Theme 3: Pacific Health Professionals’ Experiences of Providing Gout Education

Health professionals shared their experiences about providing gout education to Pacific patients, identifying challenges such as inadequate existing resources and language. Several health professionals described the current resources they use to provide gout education as too long, needing active explanations (from health professionals) and only using certain parts, such as diagrams or graphs for gout education.


*“So, the resource that I have been able to use is sourced from [NZ Health Organization] but that’s quite lengthy. It’s like a booklet. I was trying to give that but then the patients just look at it. It’s really informative but I was trying to pick out the main points because I know that they probably won’t read it”*
*[P1]*.


*“It’s this resource [Gout booklet] that has a pie chart in it and I’m showing the patients trying to explain the fact that it isn’t their fault. But then you need someone to explain it. Because if that was me and they gave me the pamphlet, and I will be asking okay what does the pie chart even mean?”*
*[P6]*.

Most participants agreed that they needed to be more innovative in their educational strategies but also keep “it as simple as possible…” [P8]. Two GPs shared their experiences of providing education to gout patients:


*“I sometimes have to draw. I’m not the best artist, but I do try and draw and explain it to them, and they come to be more appreciative of that…”*
*[P11]*.


*“People quite often don’t want to do a lot of reading, and I think for males especially the ones who watch sports, I do have a little blurb that I take them through on about gout. Just like likening allopurinol to like a wide receiver and colchicine as the blockers and I do find like younger males who know of American football really kind of resonate with that and they are like oh, okay that makes sense.”*
*[P14]*.

The majority of participants (*n* = 11) also identified language as a significant challenge when providing gout education to Pacific patients. Although there was a phone translation service available, GPs indicated having to rely heavily on their ability to speak their “own Pacific native language” [P6], “other Pacific staff members at their clinic” [P12], and patients’ family members. One nurse indicated that within Pacific ethnic groups, there are diverse dialects, which provide challenges when delivering health education.


*“We have a lot of Pacific islanders who don’t speak English very well. I am pretty fortunate that I can speak Samoan so those people I am able to help get a little bit more of an understanding of what gout is”*
*[P3]*.


*“But I’m talking about those who live in the Northern groups of the Cook Islands, where their language is more like Samoan rather than Cook Island. I don’t understand when they speak. So even when they write, the alphabet is also different to our alphabet”*
*[P12]*.


*“There is a phone translating service [available] but it takes 15 minutes to get through and at that point, the appointment is done. It’s a tricky one but I kind of just utilise what I’ve got. I have tried getting them to ring their family member and they translate as well…”*
*[P8]*.

### 3.4. Theme 4: General Impressions of the Pacific Co-Designed Gout Video and Brochure

Overall, all health professionals indicated positive impressions of both the video and brochure, with some of them wanting to use them immediately. Participants shared that they liked how the gout-related information was presented, describing the material as easy to understand, short, visual, eye-catching, and colourful.


*“It’s a visual way for them to see what I’m explaining. I’m a visual person, so I like to see things, to understand what I’m what someone’s talking about. So, using pictures is amazing”*
*[P2]*.


*“I love the colour and length as well. What was that like a few minutes? Yeah, that’s enough. Longer than that and people just zone out.”*
*[P8]*.

Furthermore, health professionals mentioned that they particularly enjoyed the Pacific-specific framing of the resources, with the background music and gout concepts being explained by a “*Pacific-sounding person or character*” *[P8]*, which made it more engaging and relatable. One nurse highlighted the cultural significance of communicating in this way:


*“…that is really how we [Pacific peoples] speak to people. You know, when sometimes we’re not too formal all the time, and that is how we will deliver and how people will take the information”*
*[P12]*.

The gout-related content in the resources was also well received, with participants indicating that they liked how it addressed all the key aspects of gout, including what gout is, the main risk factors, and gout management, especially the explanations of the types of medications (painkillers versus allopurinol). One nurse appreciated how the content focussed on addressing “*the myths around gout for these areas*” *[P9]*, while a GP expressed that she loved how gout was framed in terms of its broader life impact:


*“[with gout] you’ll miss out going to gatherings, going to sports games, that’s quite important. It’s a reminder like it really does impact your life. Why else would we be trying to do this? It’s because of your life”*
*[P8]*.

### 3.5. Theme 5: Perception of the Feasibility of the Gout Video and Brochure in a Clinical Setting

#### 3.5.1. Subtheme 1: Useability and Acceptability of the Resources in a Clinical Setting

Health professionals indicated that the video and brochure can be used as supplementary resources to support them in providing education about gout to Pacific patients. Nine participants recommended playing the video to the patients in the clinic waiting area before they see the GP. Showing the video to the patients beforehand can prime them with background knowledge about gout, so that when they see their GP, the consultation is more effective and efficient.


*“I’m just thinking, that is probably a great video to put in our waiting room. We usually have 2 waiting rooms, and we have big screens up which they’re (patients) watching and that would be something that I could see being put up as well.”*
*[P12]*.


*“Some of the patients that come in I wish that our nurse can may be take a tablet out with some headphones and get the patients to watch it while they are waiting for me so by the time they come in to see me they already have something in their head about gout so I am not starting from scratch”*
*[P6]*.


*“With that one video it would have saved me so much time explaining the stuff, just saved me time over other things and I love it”*
*[P8]*.

Eight participants considered the brochure to be useful for their face-to-face consultations, where they can reference key areas of gout. Additionally, health professionals (*n* = 6) noted that the brochure can also be used to provide patients with ongoing information:


*“The brochure is a good one, because when we’re speaking to them about their doubts we can give them a handout as well, and you know, talk to them. What some of my nurses usually do is they have a handout, and they go through the points with them. You know, they point out what we’re talking about on the brochure as well”*
*[P12]*.


*“With the QR code, patients could take it home and the families could have a look too, and that will raise the awareness of gout in the family, because it is mostly genetic, so it’s likely if you got it, then your brother’s got it…”*
*[P8]*.

#### 3.5.2. Subtheme 2: Impact of the Resources as an Educational Tool

All participants perceived the educational resources to work well as effective tools for increasing Pacific patients’ understanding of gout and its management. Community nurses and pharmacists emphasised that although the brochure was more appropriate for their work settings, they felt that the video was the most effective for improving patient education about gout.


*“I feel like the video is probably more powerful than the pamphlet. So, if there’s a way to make them sit down like before they get their gout medication or in the consult room and have a watch of this video ideally that would be good”*
*[P13]*.


*“That would be quite a good thing trying to encourage them to watch this video because I feel like it gives the most benefit on gout information and the pamphlet is like a lead on to the video”*
*[P2]*.

While the video and pamphlet were effective tools, participants indicated that health professionals play a vital role in improving the effectiveness of the resources, by actively encouraging engagement and linking the brochure to the video. One participant explained the following:


*“We put the pamphlet in [the bag] with the medications for gout and I think it would be our job to say oh here it is [the pamphlet] and if we highlight it enough in our spiel about the pamphlet, they hopefully will go home and watch it”*
*[P14]*.

Four participants, mainly nurses, emphasised how the educational resources not only improved their own understanding about gout, but could also help them to better educate others. Several GPs suggested utilising the video and brochure to build the competencies and skills of junior staff members.


*“When the video says that genes are a contributing factor to gout. This is the first time for me to hear that [gene-related] about gout”*
*[P5]*.


*“The majority of the time, I’ve just thought the food that we ate contributed to why our gout urate levels are high, but I did not know that it is a gene thing that is common in Pacific people, and their weight”*
*[P4]*.


*“Like seriously, because it’s got all the basic stuff, I think it’ll be good for training staff and could be used during clinical meetings as a refresher. For example, we’ve got like, student nurses or student doctors who are coming in. They could make use of this as well for upskilling”*
*[P8]*.

#### 3.5.3. Subtheme 3: Perceptions of Improvements and Barriers in Utilising the Resources

All participants indicated that they were happy with the current state of the resources. Minor technical adjustments were suggested, such as the inclusion of subtitles and a slight reduction in the volume of the background music to improve the clarity of the speaker’s voice. The majority of participants highlighted structural barriers that may inhibit Pacific patients’ access to the educational resources. These barriers include the clinics or pharmacies not having a television in the waiting area, and a lack of digital devices such as iPads for patients to use while waiting. Other structural barriers highlighted included time constraints, which may inhibit health professionals’ ability to present the educational resources effectively to the patients.


*“I think subtitles would be very helpful especially for those patients who can’t really hear they can read off the screen during the video.”*
*[P9]*.


*“The music is a little loud. Are you able to turn the background music down a little? But that is the small critic I have for the video. Otherwise, it is a very good resource.”*
*[P4]*.


*“I guess a barrier will be if your clinic doesn’t have a TV, you can’t really play it.”*
*[P11]*.

## 4. Discussion

Despite the high burden of gout among Pacific populations, there is still a lack of understanding about gout in the community, which impacts people’s ability to navigate health systems and receive effective gout treatment [15,18]. The aim of this study was to identify and discuss the current state and perceptions of Pacific gout education within primary care in NZ, and also explore health professionals’ views on Pacific co-designed educational resources, as well as the feasibility of using these resources in clinical settings. This qualitative study presents findings from 14 Talanga interviews with health professionals working in clinics and pharmacies across Auckland, NZ that serve high numbers of Pacific patients.

Our study identified health system barriers that hinder health professionals from delivering effective gout education to Pacific patients. Consistent with previous research [30,31,32,33], we found upstream challenges such as difficulties securing appointments, long waiting times, and high costs, which inhibit Pacific gout patients from accessing treatment. Once at the clinic, health professionals also noted that Pacific patients experience further barriers such as short consultation times and a lack of resources due to gout not being prioritised by higher health authorities. A qualitative study in NZ of 10 health professionals from an Indigenous health provider reported similar findings, highlighting the unique needs and demands of the communities that they serve, which may not align with Western funding models [32]. Participants discussed that resources are generally distributed in ways that protect the income of clinics, such as achieving national health targets, which may not reflect the unique needs of communities [32]. More flexibility to distribute clinical funding based on community needs, addressing some of the structural barriers, could create a safe environment that fosters vā, a relational space between health professionals and their patients. This could strengthen trust and communication, and in turn positively impact the gout education and treatment received by Pacific patients.

Health professionals in this study noted the lack of educational gout resources, which often placed the responsibility on them to provide patient education during appointments. This was discussed as potentially problematic depending on the health professional’s knowledge of gout, and further complicated if patients experienced language barriers. A quantitative survey of 709 community health workers from China aged 18 years and older found overall low levels of gout knowledge, with a score of 58% (17.7/30) [33]. Health professionals under the age of 30 years had the lowest scores, and GPs were found to have higher gout knowledge compared to nurses and other health professionals [33]. Further training for health professionals, focused on priority health issues like gout, could improve knowledge, and in turn ensure that accurate information is passed onto patients. The provision of patient education resources could also be a key step in reinforcing these concepts, even after clinical appointments end. Ensuring that these resources are available in different Pacific languages would also be important, in order to support better understanding for diverse population groups [30].

Health professionals in this study discussed how time constraints on primary care appointments limited their ability to deliver effective education, and thus resulted in Pacific patients seeking alternative information sources. Similar to previous research [11,12], participants in this study reported that while social media platforms and online sources can provide vast amounts of information on gout for patients, this may not always align with scientific guidelines. Health professionals particularly expressed concerns about the rise of social media’s influence on discussing herbal treatments, which can impact patients’ perceptions of long-term gout management and treatment. A content analysis of 130 YouTube videos providing gout dietary recommendations revealed that the content poorly aligned with evidence-based guidelines, with an average compliance score of only 27% [12]. Another study [34] found that alternative treatments promoted on media platforms typically did not report clinical trial quality or validity, leading to misunderstandings around effectiveness. This suggests the need for more standardised quality reporting and evidence-based and culturally appropriate gout educational resources that can address misleading information and better support Pacific communities in managing their gout.

After viewing the video and brochure co-designed by Pacific communities for Pacific gout patients, health professionals in this study responded positively, highlighting the benefits of accurate and culturally appropriate resources. Health professionals attributed the educational resources’ impact to their simplicity, Pacific cultural framing, emphasis on the broader life impacts of gout, and engaging visuals and colours. A previous mixed-methods study of 50 Marshallese participants reported that a YouTube video developed using community-based participatory research principles improved knowledge, supported health management behaviours, and increased treatment adherence [35]. The incorporation of Pacific communities’ voices in the development of the resources for the present study enabled a holistic portrayal of gout, extending beyond clinical symptoms to include wider social and personal impacts, making it more relatable and meaningful for patients.

Consistent with previous research [36,37], some study participants, particularly nurses, acknowledged their limited knowledge on gout. The co-designed video and brochure were therefore seen not only as appropriate resources for Pacific patients, but also a tool to support health professionals’ own understanding when delivering patient education. A qualitative study from England of 18 health professionals indicated that the lack of knowledge of nurses could be attributed to only treating gout as a comorbidity in patients with diabetes or renal disease accessing nurse-led clinics [36]. A qualitative study in NZ of 69 gout patients aged between 28 and 87 years reported that while some health professionals provided helpful information about gout during consultations, others delivered very little or none [37]. Additionally, some patients expressed confusion when information in the gout pamphlets contradicted what was recommended to them by their GPs [37]. This suggests the need for standardised and comprehensive gout education across all levels of healthcare providers to ensure that staff are equipped with accurate and consistent information to support building trust with patients.

Health professionals in this study discussed that to be effective, the gout resources needed to be used strategically within clinics, and that health professionals would play a vital role in incorporating them into patient care. For example, one suggestion was to play the Pacific gout video in clinics’ waiting areas to brief gout patients before seeing their doctor. On the other hand, the brochure was identified as important for supporting face-to-face patient education by reinforcing key messages. The QR code on the brochure could also link patients to the video to develop a more in-depth understanding and spread awareness at home. This was consistent with previous studies, which found that although both videos and brochures could be effective in improving disease-related knowledge, videos were more effective in enhancing immediate understanding, while brochures better maintained long-term retention [38,39,40].

This research helps to combat to the dearth of literature on the views of health professionals on co-designed education resources. A strength is that our participants included a range of health professionals, such as GPs, practice nurses, and community pharmacists, from Samoan, Cook Island Māori, and Tongan backgrounds who see high numbers of Pacific patients in their practices. Although the sample size of health professionals was perhaps small (*n* = 14), the research team agreed that data saturation was reached, and no new information was being gathered. A limitation is that other general practice-based health professionals such as health coaches and health improvement practitioners who may also provide gout education to patients were not included in the study.

Future health research may follow the co-design model used in this research, enabling communities and health professionals to address other relevant health issues that they identify and co-create interventions. The video and brochure produced from this research shall be disseminated to communities, general practices, and health websites to spread awareness and support Pacific peoples’ understanding of gout. Policymakers should prioritise the needs of communities when allocating health resources, particularly in health issues that the communities themselves prioritise. This is to ensure that treatment and management strategies are relevant, effective, and impactful.

## 5. Conclusions

This qualitative study highlighted the structural barriers that inhibit effective gout education for Pacific patients, such as limited consultation times, a lack of culturally appropriate resources, and inconsistent gout-related messaging from health professionals. The positive reception of the gout educational resources reinforces the importance of including Pacific communities in decision-making levels when designing and developing interventions to address health inequities. This will address key barriers for patients that are important for building trust and understanding to improve long-term gout management. Clinical standardised training on gout management and culturally safe communication are needed to ensure consistent and accurate education for Pacific patients.

## Figures and Tables

**Table 1 healthcare-13-02089-t001:** Characteristics of health professionals who participated in the interviews (*n* = 14).

Participant	Ethnicity	Sex	Age in Years	Occupation	Years of Experience
P1	Samoan	Female	40–49	Registered nurse	12
P2	Samoan	Female	30–39	Nurse	3
P3	Samoan	Male	30–39	General practitioner	5
P4	Tongan	Female	50–59	Community nurse	>20
P5	Tongan	Female	50–59	Registered nurse	>20
P6	Tongan	Female	30–39	General practitioner	6
P7	Tongan	Female	20–29	Registered nurse	2
P8	Tongan	Female	30–39	General practitioner	3
P9	Tongan	Male	30–39	Nurse prescriber	12
P10	Cook Islands Māori	Male	30–39	Community pharmacist	8
P11	Cook Islands Māori	Female	40–49	General practitioner	5
P12	Cook Islands Māori	Female	30–39	Nurse	9
P13	Cook Islands Māori	Male	30–39	General practitioner	4
P14	NZ European	Male	20–29	Community pharmacist	3

## Data Availability

This study has no additional data available.

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
