# Peer review of "Health Professionals’ Perceptions of Pacific Co-Designed Resources for Pacific Gout Patients"

_healthcare, 2025, doi:10.3390/healthcare13172089_

Round 1
Reviewer 1 Report
Comments and Suggestions for Authors
Comments to the Authors
Thank you for your work on this study. The research presented in this manuscript provides timely and salient information that can be utilised by New Zealand healthcare practitioners to benefit the health and wellbeing of Pacific People with gout in New Zealand. The manuscript is well-written; however, a major revision is required. Based on the study findings, I suggest that the study aim needs to be revised/modified and a major revision of the Methods section is required. The Results section needs to be formatted for clearer identification of themes, subthemes and participant quotes.
Edits
- Based on the study findings (the first three themes that were identified and discussed in the Results section), the study aim needs to be revised/modified. I think there should be two parts to the study aim. The study aims could read something like this: The present study had two main aims: (1) To identify and discuss the current state and views on Pacific patient gout education within primary care practice in New Zealand, and (2) (what is currently stated as the study aim)
- The Abstract will need to be updated based on the revised study aim(s).
- Materials and Methods section. Information needs to be provided regarding the interview guide. I suggest a separate heading for this, i.e., Interview guide. The current version of the manuscript states that the interview guide comprised of four key areas. Please provide information regarding the questions that were asked within these four main focus areas. Also, information needs to be provided regarding how the interview guide was developed. Were questions composed based on previous research findings?
- Data analysis section. It’s stated that the data analysis process comprised of six main steps. Can you please provide examples of what each step entailed. Did one researcher initially identify the themes? How were themes verified by co-authors to reduce individual researcher bias and reach consensus on the identification of each theme identifed?
- Results section. Page 4, lines 155-159, please place a number in brackets before each theme is named. For example (1) health system barriers inhibiting quality gout education.
- Each theme needs to be clearly stated in the Results section and by its full name. For example, Theme 1: Health system barriers inhibiting quality gout education. Please number and format all themes as provided in the example above.
- Were subthemes identified under the fifth theme? For example, there is a 3.5.1, 3.5.2, 3.5.3.
- Presentation of participant quotes. I’ve not seen participant quotes presented within a theme descriptor before. Participant quotes should be placed separately from the text. The text (theme descriptor) should lead up to the participant quote to provide contextual information, but the quote itself should stand alone.
- Page 6, line 262, provides a quote (I presume, as it has quotation marks), but there is no number corresponding to a participant after the quotation marks.
- Page 7, lines 282-291 discusses a subtheme (3.5.3), however there are no participant quotes within this subtheme. A theme or subtheme needs to be accompanied by participant quotes to be substantiated.
Author Response
|
Response to Reviewer 1 Comments
|
||
|
1. Summary |
|
|
|
Thank you very much for taking the time to review this manuscript. Please find the detailed responses below and the corresponding revisions/corrections highlighted/in track changes in the re-submitted files.
|
||
|
2. Questions for General Evaluation |
Reviewer’s Evaluation |
|
|
Does the introduction provide sufficient background and include all relevant references? |
Yes
|
|
|
Are all the cited references relevant to the research? |
Not applicable – not provided |
|
|
Is the research design appropriate? |
Must be improved |
|
|
Are the methods adequately described? |
Must be improved |
|
|
Are the results clearly presented? |
Yes |
|
|
Are the conclusions supported by the results? |
Yes |
|
|
Are all figures and tables clear and well-presented? |
Yes |
|
|
3. Point-by-point response to Comments and Suggestions for Authors |
||
|
Comments 1: Thank you for your work on this study. The research presented in this manuscript provides timely and salient information that can be utilised by New Zealand healthcare practitioners to benefit the health and wellbeing of Pacific People with gout in New Zealand. The manuscript is well-written; however, a major revision is required. Based on the study findings, I suggest that the study aim needs to be revised/modified and a major revision of the Methods section is required. The Results section needs to be formatted for clearer identification of themes, subthemes and participant quotes. |
||
|
Response 1: We really appreciate your time and feedback. You have pointed out valuable areas that can improve this manuscript significantly and we have made changes accordingly. |
||
|
Comment 2: Based on the study findings (the first three themes that were identified and discussed in the Results section), the study aim needs to be revised/modified. I think there should be two parts to the study aim. The study aims could read something like this: The present study had two main aims: (1) To identify and discuss the current state and views on Pacific patient gout education within primary care practice in New Zealand, and (2) (what is currently stated as the study aim) |
||
|
Response 2: We thank the reviewer for this feedback to improve clarity and we have modified our aim to include the two parts as follows: “Therefore, the present study has two main aims: (1) to identify and discuss the current state and views on Pacific gout education within primary care practice in NZ, and (2) to explore health professional perceptions on Pacific co-designed educational resources and their useful-ness in the clinical settings.” [Lines 94-97]. These aims have also been updated when referred to later in the discussion section [Lines 387-390]. |
||
|
Comment 3: The Abstract will need to be updated based on the revised study aim(s). |
||
|
Response 3: As above, we have updated the aim in the abstract, “The aim of this study is to identify and discuss the current state and perceptions on Pacific gout education, and explore health professionals’ views on Pacific co-designed resources and their usefulness in clinical settings”. (Abstract, Lines 17-19) |
||
|
Comment 4: Data analysis section. It’s stated that the data analysis process comprised of six main steps. Can you please provide examples of what each step entailed. Did one researcher initially identify the themes? How were themes verified by co-authors to reduce individual researcher bias and reach consensus on the identification of each theme identified? |
||
|
Response 4: As suggested, we have added the following details to the analysis section, “Utilising these steps, data from all the interviews were initially coded independently by the Pacific researchers SO and ST. Then, both researchers compared the theme interpretations and coding. Any discrepancies and similarities in their analysis were discussed in a brainstorming session to gain a more holistic and in-depth understanding of the results. The themes were then presented to the remaining re-searchers in the team (MO, FG, HL, MH and RL) to review before progressing with the write-up of results” [Lines 152-158]. |
||
|
Comment 5: Results section. Page 4, lines 155-159, please place a number in brackets before each theme is named. For example (1) health system barriers inhibiting quality gout education. |
||
|
Response 5: We agree with the reviewer and have updated the text to add brackets with a number before each theme as follows: “The findings from this study revealed five main themes: (1) health system barriers inhibiting quality gout education, (2) misleading information, (3) health professionals’ experiences of providing health education, (4) general impressions of Pacific co-designed resources, and (5) perceptions on the feasibility of the resources in a clinical setting.” [Lines 172-175] |
||
|
Comment 6: Each theme needs to be clearly stated in the Results section and by its full name. For example, Theme 1: Health system barriers inhibiting quality gout education. Please number and format all themes as provided in the example above. |
||
|
Response 6: Thank you for the valuable feedback, we have modified our theme sub-titles in line with the suggestion. For example, “Theme 1: Health system barriers inhibiting gout education” [Line 176]. This formatting style has been updated throughout the results section. |
||
|
Comment 7: Were subthemes identified under the fifth theme? For example, there is a 3.5.1, 3.5.2, 3.5.3. |
||
|
Response 7: Thank you for pointing this out. We have modified the subtheme titles to more clearly articulate the different sub-themes, aligning the formatting style with your previous examples above. The changes are throughout the fifth theme, for example, “Subtheme 1: Useability and acceptability of the resources in a clinical setting’ [Line 309] |
||
|
Comment 8: Presentation of participant quotes. I’ve not seen participant quotes presented within a theme descriptor before. Participant quotes should be placed separately from the text. The text (theme descriptor) should lead up to the participant quote to provide contextual information, but the quote itself should stand alone. |
||
|
Response 8: As suggested by the reviewer, we have restructured the results section to ensure most of the quotes are standalone. A few quotes remain in-text to maintain the flow of reading and we have also added further new, more comprehensive quote examples, aligned with feedback from another reviewer. The changes can be found throughout the results section [Lines 185 to 383] |
||
|
Comment 9: Page 6, line 262, provides a quote (I presume, as it has quotation marks), but there is no number corresponding to a participant after the quotation marks. |
||
|
Response 9: As suggested, we have deleted the quotation marks from this line as follows, All participants perceived the educational resources to work well as effective tools for increasing Pacific patients’ understanding of gout and its management. [Lines 338-339] |
||
|
Comment 10: Page 7, lines 282-291 discusses a subtheme (3.5.3), however there are no participant quotes within this subtheme. A theme or subtheme needs to be accompanied by participant quotes to be substantiated. |
||
|
Response 10: Thank you for pointing this out, we have added three accompanying quotes for this paragraph as follows: · ”I think subtitles would be very helpful especially for those patients who can't really hear they can read off the screen during the video.” [P9] [Lines 379-380] · “The music is a little loud. Are you able to turn the background music down a little? But that is the small critique I have for the video. Otherwise, it is a very good resource.” [P4] [Lines 381-382] · “I guess a barrier will be if your clinic doesn’t have a TV, you can’t really play it.” [P11] [Lines 383] |
||
|
4. Response to Comments on the Quality of English Language |
||
|
Point 1: The English is fine and does not require any improvement. |
||
|
Response 1: We thank the reviewer for their time and expertise. |
||
|
5. Additional clarifications |
||
|
Not applicable |
||
Reviewer 2 Report
Comments and Suggestions for Authors
The article demonstrates health professionals’ perceptions on the Pacific co-designed resources for Pacific gout patients in practice. The information in the manuscript corresponds to the scope of the Journal. In my opinion, the article will be of particular interest to employees working with populations for whom gout is a neglected or non-communicable disease. It is important to highlight the issues of management and organisation within the healthcare system, and this manuscript can serve as an example to other authors, encouraging them to address the challenges involved in healthcare.
However, there is a need to improve the article.
If possible, please provide links to the Pacific co-designed educational resources referred to by participants in the interviews.
The informed consent statement and Institutional Review Board statement are included in the article. Information on data collection and analysis is provided in the Materials and Methods section. The method for selecting participants for interviews is described. However, it would be better to create supplementary materials containing a list of the interview questions. I also recommend to more clearly indicate the goals of the interview in the form of specific points in the Study design.
Table 1 should indicate not only the age of the participants, but also how long they have worked in healthcare. It is also necessary to indicate whether the health professionals presented have encountered patients with gout. It is also necessary to indicate in the text whether the participants have a personal interest in treating gout, for example if they or their relatives have been diagnosed with it, or if they are purely concerned about this problem from a professional perspective.
To meet the Journal's publication criteria, you need to present the results of your study in more detail. For example, you could upload full interviews or interview transcripts for each healthcare participant to the public domain and provide links to resources, or add them to the supplementary materials. Currently, the information in the results appears shortened. It would be better to support them with a summary of the interviews you describe in the 'Data collection' section.
The references correspond to the topic of the article. Self-citation is present in an acceptable volume (five references out of forty-two). References from the last five years make up less than 50% (19 sources). Therefore, I recommend supplementing the references with more recent publications.
Author Response
|
Response to Reviewer 2 Comments
|
||
|
1. Summary |
|
|
|
Thank you very much for taking the time to review this manuscript. Please find the detailed responses below and the corresponding revisions/corrections highlighted/in track changes in the resubmitted files.
|
||
|
2. Questions for General Evaluation |
Reviewer’s Evaluation |
|
|
Does the introduction provide sufficient background and include all relevant references? |
Yes
|
|
|
Are all the cited references relevant to the research? |
Not applicable – not provided |
|
|
Is the research design appropriate? |
Can be improved |
|
|
Are the methods adequately described? |
Must be improved |
|
|
Are the results clearly presented? |
Must be improved |
|
|
Are the conclusions supported by the results? |
Yes |
|
|
Are all figures and tables clear and well-presented? |
Can be improved |
|
|
3. Point-by-point response to Comments and Suggestions for Authors |
||
|
Comment 1: The article demonstrates health professionals’ perceptions on the Pacific co-designed resources for Pacific gout patients in practice. The information in the manuscript corresponds to the scope of the Journal. In my opinion, the article will be of particular interest to employees working with populations for whom gout is a neglected or non-communicable disease. It is important to highlight the issues of management and organisation within the healthcare system, and this manuscript can serve as an example to other authors, encouraging them to address the challenges involved in healthcare. However, there is a need to improve the article. |
||
|
Response 1: Thank you for your time and feedback. You have highlighted out some valuable areas that have improved this manuscript significantly. We agree with your overall comments and have made these changes throughout the manuscript. |
||
|
Comment 2: If possible, please provide links to the Pacific co-designed educational resources referred to by participants in the interviews. |
||
|
Response 2: We thank the reviewer for their comment. The qualitative research reported in this paper is currently part of a larger ongoing project to evaluate the resources. Once completed, the Pacific co-designed educational resources will be published online and readily available to all. |
||
|
Comment 3: The informed consent statement and Institutional Review Board statement are included in the article. Information on data collection and analysis is provided in the Materials and Methods section. The method for selecting participants for interviews is described. However, it would be better to create supplementary materials containing a list of the interview questions. I also recommend to more clearly indicate the goals of the interview in the form of specific points in the Study design. |
||
|
Response 3: We thank the reviewer for this recommendation and have included a supplementary document with a list of the interview questions and prompts for this research (Supplementary File S1). We have also more clearly indicated the goal of the interviews as follows: “The main goal of the interview was to enable participants to discuss their views on the current state of gout education among Pacific communities.” [Lines 136-138] |
||
|
Comments 4: Table 1 should indicate not only the age of the participants, but also how long they have worked in healthcare. It is also necessary to indicate whether the health professionals presented have encountered patients with gout. It is also necessary to indicate in the text whether the participants have a personal interest in treating gout, for example if they or their relatives have been diagnosed with it, or if they are purely concerned about this problem from a professional perspective. |
||
|
Response 4: Thank you for pointing this out. We agree with the comment and have added a new column in Table 1 with each participants’ experience [Lines 163-164]. We have also added the following text indicating participants’ interest in gout: “All participants expressed concern about gout due to witnessing its high prevalence in Pacific patients at their clinics or working environment and communities. Some health professionals also shared experiences of their family and seeing patients’ family members affected by gout.” [Lines 166-169] |
||
|
Comments 5: To meet the Journal's publication criteria, you need to present the results of your study in more detail. For example, you could upload full interviews or interview transcripts for each healthcare participant to the public domain and provide links to resources, or add them to the supplementary materials. Currently, the information in the results appears shortened. It would be better to support them with a summary of the interviews you describe in the 'Data collection' section. |
||
|
Response 5: Thank you for your valuable feedback to provide more comprehensive examples in our results section. We have updated the paper to present quotes in full. Additionally, in line with another reviewer’s suggestion, we have formatted that results section to ensure that most of the quotes are standalone, rather than embedded throughout the text. The changes are made throughout the results section [Lines 185 to 383] |
||
|
Comments 6: The references correspond to the topic of the article. Self-citation is present in an acceptable volume (five references out of forty-two). References from the last five years make up less than 50% (19 sources). Therefore, I recommend supplementing the references with more recent publications. |
||
|
Response 6: As suggested by the reviewer, we have added more recent publications to this manuscript. At present, references from the last five years now make up 63% (25 sources). Due to paucity of research related to Pacific gout, some sources we have cited are understandably older than five years. |
||
|
4. Response to Comments on the Quality of English Language |
||
|
Point 1: The English is fine and does not require any improvement. |
||
|
Response 1: We thank the reviewer for their time and expertise. |
||
|
5. Additional clarifications |
||
|
Not applicable |
||
Reviewer 3 Report
Comments and Suggestions for Authors
Thank you for the opportunity to review this qualitative study about health professionals' perceptions of the Pacific co-designed resources for Pacific gout patients, also exploring the barriers, including system and personal barriers, and social media impact on health education.
I suggest some minor corrections as follows:
The study follows the COREQ criteria. There is some paucity of data about the research team. Please provide the credentials and occupation of the researchers who conducted the interviews and state their relationship with the participants (if any).
State whether there was any dropout of the participants (or whether they were not willing to participate in the study; if this is the case, give reason why).
Abstract
- Line 19 - please write clearly there were interviews conducted with 14 participants. The way it is written now may be misinterpreted as though there were 14 different interviews.
Introduction is clearly written.
Please provide reference for educational material that is being evaluated in the study throughout the manuscript (line 89, line 227, line 248, line 348, and line 394).
Materials and Methods
A comment to improve this section is already given above (second paragraph) about paucity of data about the research team, dropout rate, and willingness to participate in the study.
Results
Line 161-162 – First sentence in 3.1 section is concluded with”(Table 1)” even though Table 1 does not support this sentence.
Discussion is well written.
Author Response
|
Response to Reviewer 3 Comments
|
||
|
1. Summary |
|
|
|
Thank you very much for taking the time to review this manuscript. Please find the detailed responses below and the corresponding revisions/corrections highlighted/in track changes in the re-submitted files.
|
||
|
2. Questions for General Evaluation |
Reviewer’s Evaluation |
|
|
Does the introduction provide sufficient background and include all relevant references? |
Yes
|
|
|
Are all the cited references relevant to the research? |
Not applicable – not provided |
|
|
Is the research design appropriate? |
Yes |
|
|
Are the methods adequately described? |
Can be improved |
|
|
Are the results clearly presented? |
Yes |
|
|
Are the conclusions supported by the results? |
Yes |
|
|
Are all figures and tables clear and well-presented? |
Yes |
|
|
3. Point-by-point response to Comments and Suggestions for Authors |
||
|
Comments 1: Thank you for the opportunity to review this qualitative study about health professionals' perceptions of the Pacific co-designed resources for Pacific gout patients, also exploring the barriers, including system and personal barriers, and social media impact on health education. I suggest some minor corrections as follows. |
||
|
Response 1: Thank you for your time and feedback. You have highlighted some valuable areas that can improve this manuscript and we have made these minor corrections accordingly. |
||
|
Comment 2: The study follows the COREQ criteria. There is some paucity of data about the research team. Please provide the credentials and occupation of the researchers who conducted the interviews and state their relationship with the participants (if any). |
||
|
Response 2: As suggested, we have provided further details of the research team by including a supplementary document with the COREQ criteria and adding credentials and occupations. This is attached as Supplementary File S2. |
||
|
Comment 3: State whether there was any dropout of the participants (or whether they were not willing to participate in the study; if this is the case, give reason why). |
||
|
Response 3: Thank you for this comment. We have edited the participants and sampling section to indicate that this study had no participant dropouts, “All participants who were invited were willing to participate resulting in no dropouts.” [Lines 120-121] |
||
|
Comments 4: Abstract • Line 19 - please write clearly there were interviews conducted with 14 participants. The way it is written now may be misinterpreted as though there were 14 different interviews. |
||
|
Response 4: We agree with this comment and have edited this line as follows: “The Fa’afaletui model was utilised to conduct semi-structured Talanga interviews with 14 health professionals in Auckland, New Zealand who work in primary care clinics” [Lines 20-22] |
||
|
Comments 5: Introduction is clearly written. |
||
|
Response 5: We thank the reviewer for their comment. |
||
|
Comments 6: Please provide reference for educational material that is being evaluated in the study throughout the manuscript (line 89, line 227, line 248, line 348, and line 394). |
||
|
Response 6: We thank the reviewer for their feedback to include a reference for the education material being evaluated. The qualitative research reported in this paper is currently part of a larger project to evaluate the resources and is still ongoing. Once completed, the Pacific co-designed educational resources will be published online and readily available to all. |
||
|
Comments 7: Results Line 161-162 – First sentence in 3.1 section is concluded with” (Table 1)” even though Table 1 does not support this sentence. |
||
|
Response 7: As suggested, we have edited this sentence and included the term ‘occupation’ to link well with Table 1 as follows: “The majority of health occupations were nurses (50%), followed by GPs (36%), and pharmacists (14%) – see Table 1.” [Lines 164-165]. |
||
|
Comments 8: Discussion is well written. |
||
|
Response 8: We thank the reviewer for their comment. |
||
|
4. Response to Comments on the Quality of English Language |
||
|
Point 1: The English is fine and does not require any improvement. |
||
|
Response 1: We thank the reviewer for their time and expertise. |
||
|
5. Additional clarifications |
||
|
Not applicable |
||
Round 2
Reviewer 1 Report
Comments and Suggestions for Authors
Thank you for the revisions made to the manuscript. The revised manuscript reads well. I am satisfied with the revised manuscript. This research will benefit Pacific gout patients and their families and will provide useful information for healthcare professionals who treat these patients.
Reviewer 2 Report
Comments and Suggestions for Authors
Thank you.
The proposed additions are completely satisfactory and bring understanding to the manuscript.
The article can be accepted.